# Enhanced on-chip phase measurement by inverse weak value amplification

Meiting Song [1], John Steinmetz[2], Yi Zhang [1], Juniyali Nauriyal[1,3], Kevin Lyons[4], Andrew N. Jordan [2,5] & Jaime Cardenas[1,2✉]

Optical interferometry plays an essential role in precision metrology such as in gravitational wave detection, gyroscopes, and environmental sensing. Weak value amplification enables reaching the shot-noise-limit of sensitivity, which is difficult for most optical sensors, by amplifying the interferometric signal without amplifying certain technical noises. We implement a generalized form of weak value amplification on an integrated photonic platform with a multi-mode interferometer. Our results pave the way for a more sensitive, robust, and compact platform for measuring phase, which can be adapted to fields such as coherent communications and the quantum domain. In this work, we show a 7 dB signal enhancement in our weak value device over a standard Mach-Zehnder interferometer with equal detected optical power, as well as frequency measurements with 2 kHz sensitivity by adding a ring resonator.

[1] The Institute of Optics, University of Rochester, Rochester, NY 14627, USA. [2] Department of Physics and Astronomy, University of Rochester, Rochester, NY 14627, USA. [3] Department of Electrical and Computer Engineering, University of Rochester, Rochester, NY 14627, USA. [4] Hoplite AI, 2 Fox Glen Ct., Clifton Park, NY 12065, USA. [5] Institute for Quantum Studies, Chapman University, Orange, CA 92866, USA. ✉email: jaime.cardenas@rochester.edu

Sensitive measurements with optical interferometry play an essential role in precision metrology such as gravitational wave detection[1,2], gyroscopes[3–6] and environmental sensing. While the classical limit for sensitivity in phase measurements is the shot noise limit, it is challenging to reach this level in practical applications. Weak value amplification[7–9] can amplify the signal without increasing detected optical power, while also suppressing noises such as time correlated and systematic noise[10–12]. This allows these systems to increase the signal-to-noise ratio and achieve shot noise limited sensitivity. Previous demonstrations of weak value amplification have used a bulky apparatus that requires precise alignment and stability. Meanwhile, integrated photonic devices are revolutionizing the fields of sensing[6,13], communications[14–16] and quantum computing[17–19]. Achieving weak value amplification with integrated photonic devices provides the merit of compactness and robustness for optical interferometric sensors, as well as improving their stability and signal-to-noise ratio.

Weak value amplification increases the sensitivity of a measurement by slightly coupling the system to a state orthogonal to its original state and taking a small subset of data mostly in the orthogonal state. For example, in a well-aligned and balanced Sagnac interferometer (Fig. 1b), the constructively interfered output contains all the light (bright port), and destructively interfered output has no light coming out (dark port). Then a spatial phase front tilt, which is introduced by a slightly tilted mirror, and a small phase difference between the two paths are applied to the interferometer. One is the target parameter to be measured and the other works as coupling to an orthogonal state. This would cause a beam, weakly shifted from its original well-aligned location, to appear in the dark port, which is the small subset of the data. The beam field profile shift, measured by the power difference between the left and right halves of the detector, reflects the target parameter and gives a better sensitivity with low

power than its standard counterparts[20]. While there is some information in the rejected port (the bright port), it is a negligible fraction of the total information[10,20,21]. As detector saturation limits the maximum signal-to-noise ratio attainable, weak value amplification provides a further increase to the signal-to-noise ratio[22]. Weak value amplification techniques with optical interferometers have demonstrated improvement on sensitivity over traditional interferometers[23] for measurements of position[8,9], phase[24,25], frequency[26] and temperature[27–29].

In this work, we use "inverse" weak value amplification to enhance the phase sensitivity of an on-chip interferometer. There are two types of weak value amplification depending on what the target parameter is. Weak value amplification (WVA) consists in measuring the spatial phase front tilt, using the known phase shift to amplify the signal[8]. On the other hand, inverse weak value amplification (IWVA) consists in measuring the phase shift with the signal amplified by the known spatial phase front tilt[24]. In the WVA regime, the measured parameter, phase front tilt, is smaller than the propagation phase shift. Meanwhile, in the IWVA regime, the propagation phase shift is smaller than the phase front tilt, which is opposite from WVA. The two operating regimes allow different applications of weak value techniques. In a waveguide interferometer, phase shift is commonly used for sensing purposes as other sensing parameters, such as temperature and features of bio samples, can be easily converted to phase shifts applied to the waveguide.

## Results

**Design of individual components**. We implement inverse weak value amplification on a photonic chip by introducing a spatial phase front tilt into an on-chip Mach-Zehnder interferometer (MZI). We build the device (Fig. 1a) on a CMOS compatible photonic platform (see Methods). The input light is first split 50/50 using a directional coupler and we apply a phase signal to

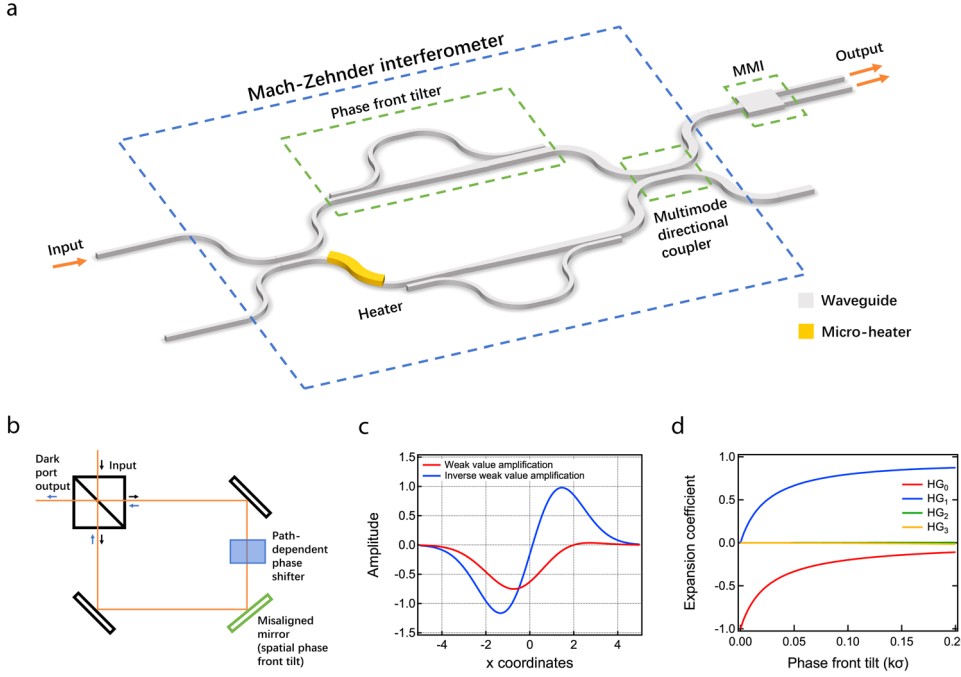

**Fig. 1 Device schematic and free space weak value amplification. a** Schematic of on-chip weak value device consisting of an MZI with phase front tilters and MMI as readouts. The heater is to apply a phase shift to the waveguides. **b** Free space Sagnac interferometer with weak value amplification. **c** Beam profile of free space dark port in weak value amplification (red) and inverse weak value amplification (blue) setup with respect to detector coordinates. In the inverse weak value regime, the beam profile resembles an $HG_1$ mode, while in the weak value regime, it resembles an $HG_0$ mode. **d** Hermite-Gaussian expansion coefficients of the inverse weak value amplification beam profile in Fig. 1c.

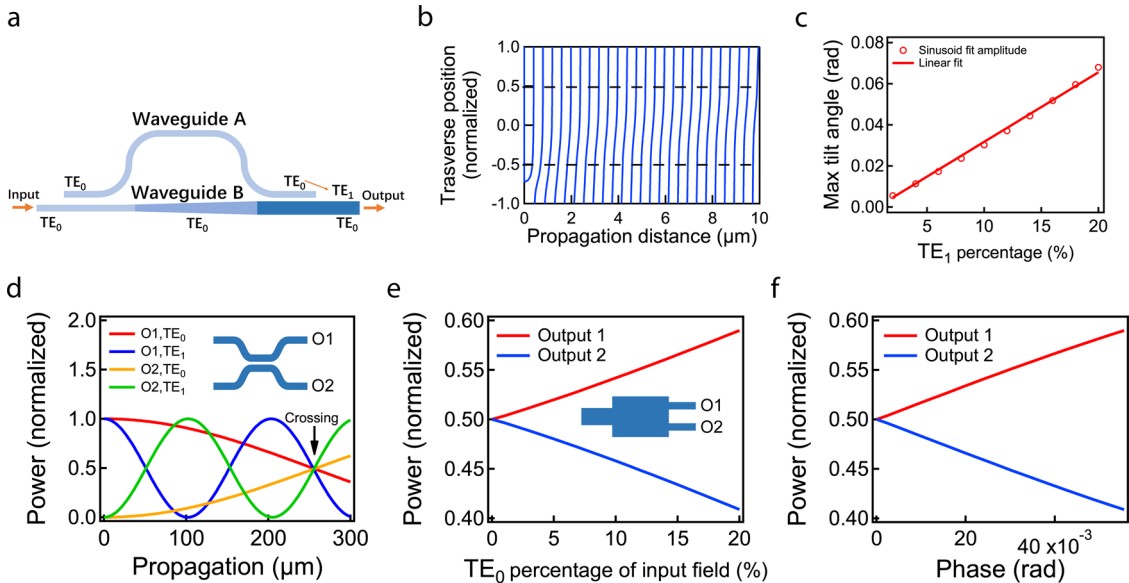

**Fig. 2 Individual components. a** Spatial phase front tilter that couples a portion of $TE_0$ mode to $TE_1$ mode to create spatial phase front tilt in waveguides. **b** Phase front contour map along propagation of a combination of $TE_0$ and $TE_1$ (7:3). The ratio is exaggerated to show the effect of the tilted phase front. **c** Phase front tilt angle vs. the field percentage of $TE_1$ mode (details in Supplementary Note 2). **d** coupler consists of two identical multimode waveguides. Simulation of power in multimode directional coupler along propagation. Coupling length is chosen to obtain a 50:50 beam splitter for both modes. **e** Multi mode interfering region with a multimode waveguide input and two single mode waveguide outputs. Multimode interfering waveguide and simulation of its output power vs. input $TE_1$ mode percentage. Power is normalized to the total power of the two outputs, which is the total power on the detector. **f** Normalized power detection vs. applied phase shift signal (details in Supplementary Note 4).

be measured on one of the arms with a micro-heater. Then light in each arm goes through a spatial phase front tilter and undergoes an opposite spatial phase front tilt. They interfere at another multi-mode 50/50 directional coupler. Finally, at the dark port, light goes through a multimode interfering (MMI) region as the readout of the spatial field profile shift. The optical power difference between the two outputs of the MMI contains the desired phase signal. The enhancement of the signal is determined by the amount of spatial phase front tilt.

To create a spatial phase front tilt in a waveguide, we use a combination of the fundamental and higher order spatial modes. We first analyze the light field in the dark port of the free space version of inverse weak value amplification with Hermite-Gaussian (HG) modes. Figure 1c shows the beam profile of the dark port in the Sagnac interferometer. Its expansion into HG modes (Fig. 1d) shows that the beam consists mostly of $HG_0$ and $HG_1$ modes in its working regime (phase front tilt larger than phase shift). The contribution of other higher order modes is negligible. Therefore, the spatial field profile shift can be considered equivalent to a combination of $HG_0$ and $HG_1$ modes and the extent of the shift is determined by the ratio between these two modes. The same applies to the spatial phase front tilt. Adding light in the $HG_1$ mode to the light in the $HG_0$ mode tilts the phase front of the beam. Applying the same theory to waveguide modes, only the zeroth and first order transverse electric ($TE_0$ and $TE_1$) modes are required for weak value amplification with integrated photonic devices. The spatial phase front tilt can be considered as coupling light from the $TE_0$ mode to the $TE_1$ mode, and the amount of coupling determines how much spatial phase front is introduced. Also, after interference, the phase shift is also determined by the ratio between $TE_0$ and $TE_1$ modes[30]. The electric field at the dark port can be expressed by the following equation, where $\phi$ is the phase difference between the two paths

and $a$ is the amplitude of the $TE_0$ mode that couples to the $TE_1$ mode (details in Supplementary Notes 1–3).

$$E_d \approx i \left[ (1-a)\frac{\phi}{2}TE_0 + aTE_1 \right] = ia \left[ TE_1 + \frac{1-a}{a}\frac{\phi}{2}TE_0 \right]. \quad (1)$$

We design a spatial phase front tilter to introduce the phase front tilt by coupling part of the $TE_0$ mode into the $TE_1$ mode. The spatial phase front tilter (Fig. 2a) starts with two identical single mode waveguides. Waveguide B couples a small portion of the light in the $TE_0$ mode of waveguide A. Then waveguide B adiabatically changes width into a multimode waveguide that supports both $TE_0$ and $TE_1$ modes. The light in the $TE_0$ mode in waveguide B stays in the $TE_0$ mode as the waveguide width changes. We design waveguide B so its $TE_1$ mode supported after the taper is phase matched to the $TE_0$ mode in waveguide A. Therefore, when waveguide A couples light back into waveguide B, the light couples into the $TE_1$ mode[16,17]. This way, we have a combination of $TE_0$ and $TE_1$ modes in the multimode waveguide (end of waveguide B), which is the equivalent of having a phase front tilt (Fig. 2b). By controlling the coupling ratio of the first directional coupler, we can adjust the amount of phase front tilt that we are introducing (Fig. 2c).

For both $TE_0$ and $TE_1$ modes to interfere, we design a directional coupler that splits 50:50 for both $TE_0$ and $TE_1$. Since the $TE_0$ mode couples with a different strength than the $TE_1$ mode, the coupling length needed for the two modes to couple half of the light into the other waveguide is usually different. However, since the coupling process is periodic, we design the gap and the length of the directional coupler so that the $TE_0$ mode goes through one quarter of a coupling cycle, while the $TE_1$ mode goes through one and a quarter coupling cycles (Fig. 2d). The result is a 50:50 directional coupler for both the $TE_0$ and $TE_1$ modes simultaneously.

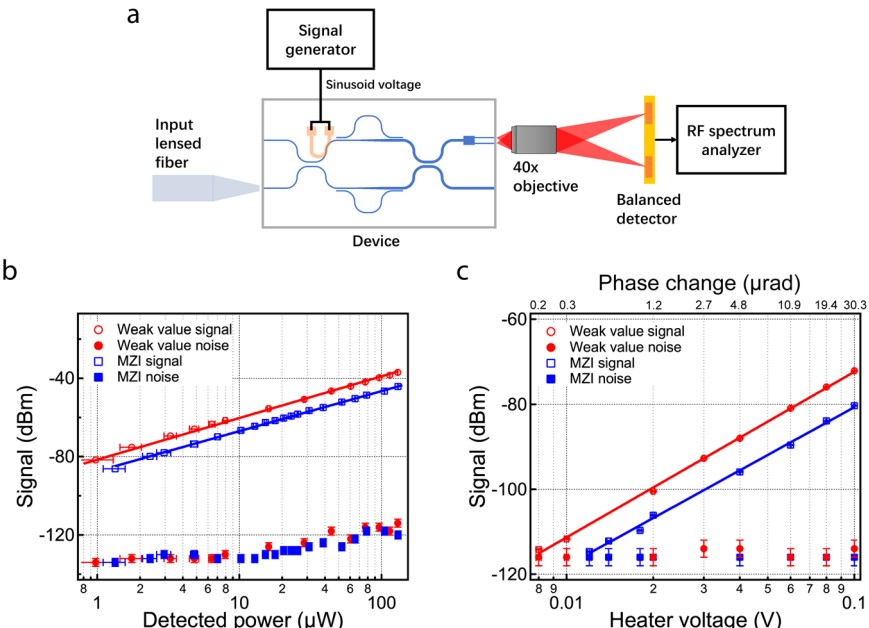

**Fig. 3 Testing and results. a** Testing setup with modulated phase signal and RF spectrum analyzer to detect corresponding detector signal. Laser light is sent in through a polarization maintaining tapered fiber. Outputs are imaged to a balanced detector with a 40x objective. **b** The signal of the weak value device and the regular MZI vs. detected power on a semi log plot. Signal and noise floor measured by RF spectrum analyzer on weak value device (red circles) and standard MZI (blue squares) with same applied phase signal and changing detected optical power. The signal is the electrical power generated by the balanced detector at a frequency of 20 kHz. The signal data are fitted to the lines. The error bars of the detected power represent the standard deviation. The error bars of the signal represent the resolution limit of the measurement. **c** The signal of the weak value device and the regular MZI vs. heater voltage and phase change on a semi log plot. Signal and noise floor on weak value device (red circles) and standard MZI (blue squares) with equal detected optical power within 5% variation and changing applied heater signal (phase signal). The signal data are fitted to the lines. The error bars of the heater voltage represent the standard deviation of the applied voltage as reported by the oscilloscope. The error bars of signal represent the resolution limit of the measurement. The top axis shows the corresponding phase shift introduced by the micro heater (details in Supplementary Note 5).

To determine the phase shift with an optical power signal, we design a multi-mode interferometer (MMI) whose power outputs are dependent on the ratio between the input $TE_0$ and $TE_1$ modes. We simulate an MMI with a multimode waveguide input and two single mode outputs with the rigorous Eigen Mode Expansion (EME) method (FIMMPROP, Photon Design). The power in the two outputs of the MMI is linear with respect to the ratio of the $TE_0$ and $TE_1$ modes in the input waveguide (Fig. 2e). As the phase signal applied to the weak value device increases, the amount of the $TE_0$ in the dark port also increases, causing the difference of the power between the two outputs of the MMI to increase as well (Fig. 2f). From the power difference, we can get the amplitude ratio $TE_0 : TE_1 = p/(1-p)$, where $p = 1.11\Delta I_{wv}$ for this design (details in Supplementary Note 4). $\Delta I_{wv}$ is the power difference normalized to the total power in the two output waveguides. Then the phase signal can be obtained by (details in Supplementary Note 4)

$$\phi = 2\frac{p}{1-p}\frac{a}{1-a}. \tag{2}$$

**Testing and comparison to standard MZI.** We test the device by applying a phase signal to the microheaters and compare its performance with a standard on-chip Mach-Zehnder interferometer. Silicon nitride has a thermo-optic effect of $2.45*10^{-5}/°C$[31]. As we apply voltage on the microheaters above the waveguide, the temperature of the waveguide increases, which changes the refractive index of silicon nitride. Therefore, the phase accumulated by the mode also changes, which serves as the target parameter to be measured. Since the phase signal applied is weak, we test the device by modulating the phase signal and detect the output signal with a RF spectrum analyzer (Fig. 3a). We apply a 10 kHz sinusoid voltage

signal of between 0 V and 1 V on the micro heater. The voltage signal creates a phase difference of about 9mrad between the signal arm and reference arm. Since the phase changes with the applied heater power, the phase signal appears at a frequency of 20 kHz on the RF spectrum analyzer. The optical outputs of the two waveguides are imaged on to a balanced detector with a 40x objective. The difference of the optical power on the two sides of the detector is converted to a voltage signal by the detector, which is then converted to electrical power on the RF spectrum analyzer. We take the electrical power signal as our signal and noise. The signal is measured with a resolution bandwidth of 0.1 Hz. At this modulation frequency, the contribution of the 1/f noise is reduced.

To confirm the enhancement of the signal, we also test an on-chip Mach-Zehnder interferometer with the same waveguide parameters and similar footprint with the same signal and testing setup. We also take the signal as the difference of the two waveguide outputs in the standard MZI. The relation between the optical power and the phase signal is as follows, where $\Delta I$ is the power difference of the two outputs, and $I_0$ is the total power of the two outputs,

$$\phi \approx \sin(\phi) = \frac{\Delta I}{I_0} \equiv \Delta I_{MZI}. \tag{3}$$

Comparing the power difference from the two devices with the same applied phase shift, we derive the amplification factor (details in Supplementary Note 4)

$$\frac{\Delta I_{wv}}{\Delta I_{MZI}} = \frac{0.45(1-a)}{a}. \tag{4}$$

In our design, $a$ is 0.125, which corresponds to a 9.97 dB amplification of the signal. The standard MZI has the same arm length as the weak value device, except that one arm is shorter to

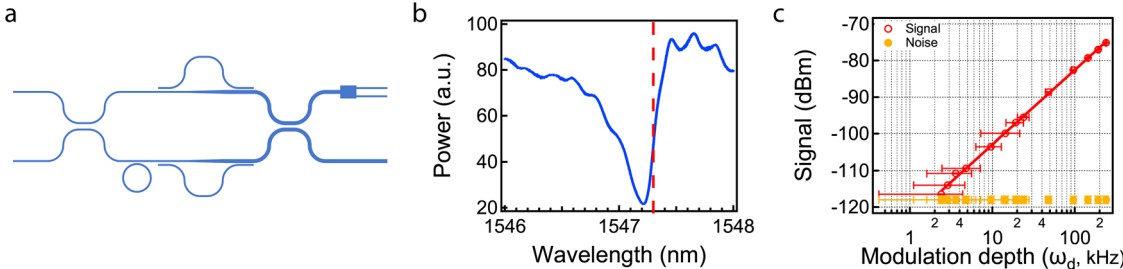

**Fig. 4 Frequency measurements. a** Schematic of inverse weak value device with ring resonator for frequency shift measurements. **b** Resonance of the ring resonator. The dashed line indicates the center of laser frequency modulation. **c** Signal and noise floor of the inverse weak value device with changing optical frequency modulation depth. The signal data are fitted to the lines. The error bars of the modulation depth represent the standard deviation of the applied voltage as reported by the oscilloscope. The error bars of the signal represent the resolution limit of the measurement.

put the MZI in quadrature. The waveguide and the heater in the MZI have the same dimensions as the single mode waveguide in the weak value device. Both heaters, including the contact probe and wires, have a resistance of $746 \pm 6\,\Omega$. This guarantees that the same phase shift, within the error range, is applied to both devices when applying the same voltage to the heaters. We also test the standard MZI with the same setup as the weak value devices. We apply the same 10 kHz sinusoid voltage wave to the heater and take the difference of the two outputs of the standard MZI with the balanced detector. Then we measure the signal with the spectrum analyzer using the same settings (100 Hz span, 0.1 Hz bandwidth). Since we are comparing the signals of our device and the regular MZI with equal detected optical power, the signals are not normalized.

We show a 7 dB signal enhancement with the inverse weak value amplification MZI when compared to a standard MZI. We take the limiting resource as the maximum optical power that can be detected. To make the comparison, we first match the detected power of the two devices. While the signal of the weak value device is amplified by 7 dB, the noise floor is not amplified (Fig. 3b). Therefore, the signal-to-noise ratio is also increased by 7 dB. The signal enhancement stays the same as the detected optical power changes. For a detected power of 130 µW, the weak value device has an input laser power of 16 mW and the standard MZI has an input laser power of 0.5 mW. The noise floors are measured with the RF spectrum analyzer at 20 kHz over a span of 0.1 kHz. They are the same for both devices because the same amount of optical power is received by the detector. At lower optical power levels ($<20\,\mu$W), the noise floor is limited by the built-in electronic amplifier noise and other electronic noise of the balanced detector. For detected power above 20 µW, the noise floors of both devices start to grow linearly with optical power in the log scale plot, which indicates that the systems start to be shot noise limited.

The weak value device shows a lower minimum resolvable signal than the standard MZI. We maximize the detected optical power ($0.22 \pm 0.01$ mW in total) in both devices and lower the applied heater voltage to compare them (Fig. 3c, resolution bandwidth of 0.1 Hz). The minimum resolvable signal is obtained by extrapolating the fitted curve of the signal and the noise floor until they cross. The standard MZI shows a minimum resolvable signal of 0.012 V, which corresponds to a phase signal of 0.44 µrad (details in Supplementary Note 5). The minimum resolvable signal of the weak value device is 0.008 V, which corresponds to a phase signal of 0.2 µrad. In situations limited by detector saturation, this technique can enhance the signal without increasing the optical power on the detector, therefore resulting in a higher signal-to-noise ratio for equal detected power.

**Frequency shift measurements**. We demonstrate laser frequency shift measurements with inverse weak value amplification by adding a ring resonator to convert changes in optical frequency into changes in phase. We replace the microheaters with a ring resonator (Fig. 4a), which adds a phase signal depending on the optical frequency. We tune the laser wavelength on the slope of the resonance (Fig. 4b, dashed line) and modulate the optical frequency with a sinusoid voltage signal of 20 kHz ($\omega_m$). The amplitude of the voltage signal determines modulation depth of the optical frequency ($\omega_d$). The laser frequency ($\omega_{laser}$) oscillates around the center frequency ($\omega_0$) in the following manner:

$$\omega_{laser} = \omega_0 + \omega_d * \sin(\omega_m t). \qquad (5)$$

Then we measure the output signal with the setup in Fig. 3a. The maximum modulation depth $\omega_d$ we apply on the optical frequency is 250 kHz, which corresponds to 15 fm of wavelength change. Such a laser wavelength change has negligible effect on the performance of the integrated photonic components, including the ring resonator (FWHW = 2.5 nm). The signal power increases linearly with modulation depth (Fig. 4c) showing a signal to noise ratio of 25 dB for 100 kHz optical frequency shifts. By extrapolating the fitting curves of the signal and the noise floor till they cross, we obtain the optical frequency detection limit of $\omega_d = 2$ kHz. This precision corresponds to measuring an optical frequency shift of 1 part in $10^{11}$. The quality factor of the ring resonator is about 9000, which is limited by radiation losses of the mode. The phase has a linear relationship with the ring's quality factor (details in Supplementary Note 6). The device is capable of 1 Hz sensitivity with a $2*10^7$ quality factor, which has been previously demonstrated in a silicon nitride photonic platform[32]. Previous free space frequency measurements with inverse weak value amplification show a 129 kHz/$\sqrt{}$Hz sensitivity (i.e. an optical frequency shift of 129 kHz could be measured with an integration time of 1 s and an SNR of 1)[26]. Taking into consideration the resolution bandwidth of 0.1 Hz, our 2 kHz detection limit corresponds to a sensitivity of 6.3 kHz/$\sqrt{}$Hz.

Integrated optics can be readily adapted to the fully quantum optical domain[33]. Quantum sensors can further leverage quantum advantage using squeezed or entangled photons, to enable devices such as quantum gyros[34]. Weak value amplification has been demonstrated for fully quantum systems in past work, both in optical[35,36] and solid state systems[37,38]. We have demonstrated in this article how the concentration of information about the parameter of interest into a smaller post-selected fraction can greatly enhance the precision of measurements given a finite power limitation of the detector by permitting more input photons[22]. This feature can be carried over into the quantum realm to either improve the post-selection fraction, or the degree

of amplification[39], whilst preserving the quantum-enhanced Heisenberg scaling of the precision. Our current demonstration of weak value amplification in an integrated optical chip paves the way to incorporate weak value based techniques for quantum optical technologies in this scalable and robust platform. The integrated weak value platform enhances classical interferometric sensing by producing a stronger signal and a higher signal-to-noise ratio than standard interferometric techniques.

## Methods

**Micro-fabrication.** We fabricate the device with CMOS compatible process. The waveguide (300 nm thick, 1.05 μm wide for single mode and 2.5 μm for multi-mode) consists of LPCVD (low pressure chemical vapor deposition) silicon nitride with 4 μm thermal silicon dioxide on bottom and 3 μm PECVD (plasma enhanced chemical vapor deposition) silicon dioxide cladding on top. We pattern the waveguides with e-beam lithography and etch with ICP-RIE (inductively coupled plasma reactive ion etcher). Finally, we sputter and lift-off platinum as micro heaters (100 nm thick, 3 μm wide, 150 μm long).

**Waveguide simulations.** Waveguide simulations are done with FIMMWAVE and FIMMPROP by Photon Design.

## Data availability

The datasets that support this study are available from the corresponding author on reasonable request.

## Code availability

The codes used for analysis and simulations are available from the corresponding author on reasonable request.

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

## Acknowledgements

The authors acknowledge funding from Leonardo DRS and A. N. Jordan Scientific. This research was funded in part by CEIS, an Empire State Development-designated Center for Advanced Technology. This work was performed in part at the Cornell NanoScale Facility, an NNCI member supported by NSF Grant NNCI-2025233. We also thank Marco Lopez, Avik Dutt, John C. Howell, Matthew T. Moen and Benjamin L Miller for discussions of the project.

## Author contributions

J.S., M.S., J.C., K.L. and A.N.J. conceived and developed the theory of weak value amplification with waveguides. M.S. and J.C. conceived and designed the integrated photonic devices. M.S., Y.Z., and J.N. fabricated the devices. M.S. and J.C. conceived, designed and carried out the experiments and analyzed the data. M.S., Y.Z., J.N., A.N.J. and J.C. contributed to writing the manuscript.

## Competing interests

This work was supported by Leonardo DRS. DRS supported the development of on-chip weak value amplification, but played no further role in the conceptualization, design, data collection, analysis, decision to publish, or preparation of the manuscript. A.N.J. discloses part of this work was carried out by his LLC outside of the University of Rochester. The remaining authors declare no competing interests.
