## [Peer Review File · Nature Communications]

REVIEWER COMMENTS

Reviewer #1 (Remarks to the Author):

This manuscript describes the first implementation of weak value amplification (WVA) on-chip. As such, it opens a wide range of applications for WVA. Weak value amplification is a general method to multiply the shift induced by some phenomena that is being measured. The expense is that the multiplication fails in most trials (or, equivalently, for most photons). Even with this failure rate, multiple groups have shown that the overall sensitivity of WVA is almost equal to that without WVA. Moreover, in the presence of some types of technical noise, WVA can outperform conventional measurement. Technical noise is the limiting factor for almost all commercial sensors. So, WVA holds much promise for improving sensors but little progress has been made beyond proof-of-principle. This manuscript takes a big step toward demonstrating that WVA can improve real commercial sensors.

The device in the paper measures phase. But phase, in turn, is changed by many environmental factors such as temperature, surrounding gases, strain, etc. Indeed, for a second device the manuscript describes the measurement of a frequency shift. In summary, the aim and the results, if correct, of the manuscript are very significant and will have both a short and long-term impact on science and industry. Based on that, it should be published in Nature Communications.

However, I have reservations about the interpretation of what the authors have shown experimentally that I would like to hear back about. These reservations centre on a lack of clarity in the manuscript about sensitivity and improvement in sensitivity. The paper expends most of its text describing the device rather than the metrological theory (and rigour) behind the device. I suggest rebalancing the content somewhat. Specifically, the paper must define what it means by sensitivity, preferably with a formula. It must also clearly define what is the constrained resource (a key part of any metrological analysis)? Is it the power at entrance of the chip, the power at the detector, the power passing through the phase-shifter? At the moment, it is unclear. There are mentions of normalizing to the power at the balanced detector. Is this then the constraint?

Specific questions/suggestions:

1. Line 51 needs at least one citation to support its claim that WVA "gives better sensitivity..."
2. I got a bit confused about the connection between tilting a beam (via a mirror) and the phase front tilt. The authors might want to say explicitly that these are the same thing.
3. The sentences on lines 57-58 are missing words. But even with those added, I think this could be worded more clearly.
4. Why doesn't the horizontal axis in Fig. 1d go to zero? Measuring near-zero tilts was the goal in the Sagnac.
5. In Fig. 1 c), the beam profile is very distorted compared to the input beam (a Gaussian). Does that mean that the experiment is operating outside the weak-value regime? If so, is that important?
6. In line 76, it says the optical power difference is the signal for WVA. Is this normalized to the total power at the detector or not. This would be much clearer with some formulas and parameter symbols.
7. Line 143, the main body of the paper needs more detail about how the two devices, MZI and WVA are equivalent. This is in the SI but belongs here since the paper hinges on it.
8. Fig. 3 refers to signals for the MZI and WVA but up to this point these signals have not been explicitly defined.
9. The authors should comment on the source of the noise in the system and its characteristics.
10. Fig 3 c. A horizontal axis for phase would be clearer than voltage.
11. Lines 159 to 165 start discussing results for "sensitivity" but it is not clear how they were found from the experiments nor defined theoretically.
12. Does the speed of modulation of the phase affect anything (e.g. the amount of noise)?
13. The frequency modulation is confusing since there is now a modulation frequency and an optical frequency. E.g the sensitivity is stated as 2KHz, but of which frequency? The reader might find that confusing. Moreover, they are of the same size. Do the generated frequency sidebands from the sinusoidal modulation affect anything?
14. Does the frequency shift consequentially alter the operation of the e.g. the MMI? If not, this should be stated with some brief justification.
15. In the SI, eq 12 is incorrect. I_0 should also be defined.
16. The SI is still not clear about the normalization in the WVA signal. Please define it with a formula.
17. The SI should contain some theoretical justification for why the WVA will outperform the MZI. As a reminder, in the absence of technical noise, the two methods have almost equal sensitivity. If it is technical

noise, the paper should explain the reason WVA outperforms the MZI for the type of technical noise observed for the chip should explained

Reviewer #2 (Remarks to the Author):

This work by Song et al develops a design to demonstrate weak value amplification using an integrated photonic chip, and then proceeds to presents data from an experimental implementation. The key challenge in moving from an optical table setting to the chip implementation is the realization of a phase tilt, which is realised by splitting - and then recombining - a waveguide mode (in analogy to a free space Hermite-Gaussian mode decomposition). This design step is supported by appropriate modelling. The resulting fabricated device is then tested and compared against a "standard" MZ interferometer in two configurations, one measuring small phase shifts induced by a micro-heater, the other measuring a frequency shift that is converted to a phase shift via a coupled ring resonator. In both cases the weak value amplified device displays significantly improved performance as expected.

The approach, implementation and results all seem sound, technically valid and and convincing to me. Further, I believe this work is original and represents an advance in the translation of laboratory experiments towards practical technologies. In this regard, I am satisfied the reported work is sound and of a publishable standard. However, focusing on the translation of a well-known and well understood effect to a different platform, I am not sure whether the present manuscript offers quite enough novelty and innovation to influence thinking in the field. Moreover, in my opinion the presentational standard of the current manuscript needs some work, for several reasons listed below and as follows: probably in an attempt to make it suitable for a broad readership, it provides a verbal and non-technical description which in my opinion lacks depth even for the non-specialist reader. In fact, the absence of equations and a more quantitative description makes it hard to follow in detail and appreciate the work fully for the expert reader as well.

Some specific questions and comments follow below:

- The authors write about it being desirable to operate at the shot noise limit as a major motivation for weak value amplification. While it is true this represents the ultimate limit of sensitivity on a per photon basis, it is not immediately obvious to me that working in this limit of low signal intensity is always desirable, and perhaps this should be discussed and explained in more detail.
- Weak value amplification can certainly be valuable and useful, as demonstrated by many impressive experimental results over the years. However, it comes at a cost of low post-selection probability, and there are basic arguments that post-selection should never be useful at a fundamental level: why not simply keep and process all the data? I recall an interesting debate from a while ago about whether and under what conditions weak value amplification helps. I appreciate the authors may consider this a bit peripheral and may not wish to discuss this in too much detail (and I agree the specific advantages they describe all seem valid), however, for a more balanced introduction, I would nonetheless have expected some degree of coverage of these issues.
- I found the sentence spanning lines 57-59 hard to read, this could be rephrased and possibly expanded upon.
- There seems to be a grammar issue in the sentence spanning lines 73-74, possibly just a 'to' missing.
- In Fig.1, I'm not sure how helpful it is to invoke the Sagnac interferometer in (b), seeing the implemented device is a MZ, as shown in (a). I appreciate the Sagnac example forms the basis of discussion in the introduction, but should this not ideally be done with reference to an the MZ setup to help the reader understand the experimental device?
- The quality and polish of the figures could be improved in terms of appearance and resolution.
- The Supplementary Materials document contains a number of typos and would benefit from thorough proofreading. In my view, it is not possible to properly understand the phase tilt implementation from the main text without reading the helpful and fairly easy to follow derivation in the SM. I am not sure if the split between those documents is quite ideal.
- The "standard" MZ benchmark seems fair and carefully set up. To assess the potential and usefulness of this approach for practical applications, some level of discussion of how this integrated WVA compares to the performance of possible competing approaches more widely than the implemented MZI would be valuable. While admittedly an apples to oranges comparison, I would also find it interesting how it compares to best-of-class / typical table top interferometers, should the authors have relevant data available.

In summary, at heart the work described in this paper presents a promising engineering approach for implementing weak-value-based phase estimation on chip, demonstrating potential for development into a

practical technology. However, as the effect described is already well understood from the laboratory setting, I do not think this manuscript offers much in terms of providing new knowledge, novel insights or a deeper physical understanding.

Reply to reviewers:

Reviewer #1 (Remarks to the Author):

We thank the referee for the detailed and helpful comments of our manuscript. We reply in detail below:

“However, I have reservations about the interpretation of what the authors have shown experimentally that I would like to hear back about. These reservations centre on a lack of clarity in the manuscript about sensitivity and improvement in sensitivity. The paper expends most of its text describing the device rather than the metrological theory (and rigour) behind the device. I suggest rebalancing the content somewhat. Specifically, the paper must define what it means by sensitivity, preferably with a formula.”

We added description and equations on lines 96-99, lines 137-141, and lines 160-166 to better describe the theory of our weak value device and how it compares to the regular MZI device.

“The electric field at the dark port can be expressed by the following equation, where ϕ is the phase difference between the two paths and a is the amplitude of the TE_0 mode that couples to the TE_1 mode (details in supplementary information).

$$E_d \approx i \left[(1-a) \frac{\phi}{2} TE_0 + a TE_1 \right] = ia \left[TE_1 + \frac{1-a}{a} \frac{\phi}{2} TE_0 \right].”$$

“From the power difference, we can get the amplitude ratio $TE_0:TE_1 = p/(1-p)$, where $p = 1.11\Delta I_{wv}$ for this design (details in supplementary information). ΔI_{wv} is the power difference normalized to the total power in the two output waveguides. Then the phase signal can be obtained by (details in supplementary information)

$$\phi = 2 \frac{p}{1-p} \frac{a}{1-a}.”$$

“The relation between the optical power and the phase signal is as follows, where ΔI is the power difference of the two outputs, and I_0 is the total power of the two outputs.

$$\phi \approx \sin(\phi) = \frac{\Delta I}{I_0} \equiv \Delta I_{MZI} .$$

Comparing the power difference from the two devices with same applied phase shift, we derive the amplification factor (details in supplementary information)

$$\frac{\Delta I_{wv}}{\Delta I_{MZI}} = \frac{0.45(1-a)}{a} .”$$

We also replaced the term “sensitivity” with “minimum resolvable signal” in the paragraph starting from line 202, which better describes our data. We take the phase shift we can measure with a SNR of 1 as the minimum resolvable signal.

“The weak value device shows a lower minimum resolvable signal than the standard MZI. We maximize the detected optical power (0.22 ± 0.01 mW in total) in both devices and lower the applied heater voltage to compare them (Fig.3c, resolution bandwidth of 0.1Hz). The minimum resolvable signal is obtained by extrapolating the fitted curve of the signal and the noise floor till they cross. The standard MZI shows a minimum resolvable signal of 0.02V, which corresponds to a phase signal of 1.2 μ rad (details in supplementary information). The minimum resolvable signal of the weak value device is 0.008V, which corresponds to a phase signal of 0.2 μ rad.”

We added the following sentences on lines 152-157, lines 196-201, and lines 204-205 to clarify our definition of the signal, the noise floor, and the minimum resolvable signal.

“The optical outputs of the two waveguides are imaged on to a balanced detector with a 40x objective. The difference of the optical power on the two sides of the detector is converted to a voltage signal by the

detector, which is then converted to electrical power on the RF spectrum analyzer. We take the electrical power signal as our signal and noise. The signal is measured with a resolution bandwidth of 0.1Hz.”

“The noise floors are measured with the RF spectrum analyzer at 20kHz over a span of 0.1kHz. They are the same for both devices because the same amount of optical power is received by the detector. At lower optical power levels (<20μW), the noise floor is limited by the built-in electronic amplifier noise and other electronic noise of the balanced detector. For detected power above 20μW, the noise floor starts to grow with optical power, which indicates that the system starts to be shot noise limited.”

“The minimum resolvable signal is obtained by extrapolating the fitted curve of the signal and the noise floor till they cross.”

“It must also clearly define what is the constrained resource (a key part of any metrological analysis)? Is it the power at entrance of the chip, the power at the detector, the power passing through the phase-shifter?”

The constrained resource is the power at the detector. We added the sentence below on line 192 to clarify this.

“We take the limiting resource as the maximum optical power that can be detected.”

Because while the laser power can be readily increased in our system, most detectors saturate around 1mW.

“At the moment, it is unclear. There are mentions of normalizing to the power at the balanced detector. Is this then the constraint?”

The constraint is the detected power on the balanced detector.

In the calculation and simulation of the devices, we normalized the power to the total detected power on the balanced detector. However, for the testing results, the signal is not normalized to the detected power because we are comparing the signals with equal detected power.

We specified on lines 137-140 that the simulation is normalized:

“From the power difference, we can get the amplitude ratio $TE_0:TE_1 = p/(1 - p)$, where $p = 1.11\Delta I_{wv}$ for this design (details in supplementary information). ΔI_{wv} is the power difference normalized to the total power in the two output waveguides.”

And we added on lines 176-177 that the testing results are not normalized:

“Since we are comparing the signals of our device and the regular MZI with equal detected optical power, the signals are not normalized.”

Specific questions/suggestions:

1. Line 51 needs at least one citation to supports its claim that WVA “gives better sensitivity...”

We added reference 20 on line 44, which demonstrate better sensitivity than traditional interferometric methods.

“...gives a better sensitivity with low power than its standard counterparts²⁰.”

20. Viza, G. I., Martínez-Rincón, J., Alves, G. B., Jordan, A. N. & Howell, J. C. Experimentally quantifying the advantages of weak-value-based metrology. *Phys. Rev. A* **92**, 032127 (2015).

2. *I got a bit confused about the connection between tilting a beam (via a mirror) and the phase front tilt. The authors might want to say explicitly that these are the same thing.*

We modified the sentence on lines 38-39 specifying that the spatial phase front tilt is introduced by the mirror tilt.

“Then a spatial phase front tilt, which is introduced by a slightly tilted mirror, and a small phase difference between the two paths are applied to the interferometer.”

3. *The sentences on lines 57-58 are missing words. But even with those added, I think this could be worded more clearly.*

We modified the sentences on lines 54-60 to make them clearer.

“Weak value amplification (WVA) consists in measuring the spatial phase front tilt, using the known phase shift to amplify the signal⁸. On the other hand, inverse weak value amplification (IWVA) consists in measuring the phase shift with the signal amplified by the known spatial phase front tilt²⁴. In the WVA regime, the measured parameter, phase front tilt, is smaller than the propagation phase shift. Meanwhile, in the IWVA regime, the propagation phase shift is smaller than the phase front tilt, which is opposite from WVA. The two operating regimes allow different applications of weak value techniques.”

4. *Why doesn't the horizontal axis in Fig. 1d go to zero? Measuring near-zero tilts was the goal in the Sagnac.*

We have changed Fig.1d to show the curve down to zero.

d Hermite-Gaussian expansion coefficients of inverse weak value amplification beam profile in Figure 1c.

The reason that the original figure does not go to zero is that this represents the regime (phase front tilt larger than 0.05) that we work with. In our implementation, we are measuring very small propagation phase shifts, using the phase front tilt to amplify the signal. We put the following text on lines 54-60

“Weak value amplification (WVA) consists in measuring the spatial phase front tilt, using the known phase shift to amplify the signal⁸. On the other hand, inverse weak value amplification (IWVA) consists in measuring the phase shift with the signal amplified by the known spatial phase front tilt²⁴. In the WVA regime, the measured parameter, phase front tilt, is smaller than the propagation phase shift. Meanwhile, in the IWVA regime, the propagation phase shift is smaller than the phase front tilt, which is opposite from WVA. The two operating regimes allow different applications of weak value techniques.”

and on lines 85-87 to clarify this point:

“Its expansion into HG modes (Fig.1d) shows that the beam consists mostly of HG₀ and HG₁ modes in its working regime (phase front tilt larger than phase shift).”

5. In Fig. 1 c), the beam profile is very distorted compared to the input beam (a Gaussian). Does that mean that the experiment is operating outside the weak-value regime? If so, is that important?

This is due to the difference between WVA and IWVA. The output beam profile of WVA consists mostly of the HG₀ mode, so it resembles a Gaussian profile. However, the output beam profile of IWVA in its working regime consists mostly of the HG₁ mode, which results in the asymmetric profile shown in Fig.1c. We added WVA beam profile in Fig.1c to compare with IWVA profile to make this difference clearer.

C

c Beam profile of dark port in weak value amplification and inverse weak value amplification setup with respect to detector coordinates. In the inverse weak value regime, the beam profile resembles an HG₁ mode, while in the weak value regime, it resembles an HG₀ mode.

6. In line 76, it says the optical power difference is the signal for WVA. Is this normalized to the total power at the detector or not. This would be much clearer with some formulas and parameter symbols.

In the simulation showing in Fig.2e, the optical power is normalized to the total power arriving at the detector. However, in the testing results, the optical power is not normalized. In order to make a fair comparison, we always consider both the weak value device and the MZI experiments where the detected powers are the same.

We added the sentences below on lines 139-140 and lines 160-163 to explain how the simulation results are normalized.

“ ΔI_{wv} is the power difference normalized to the total power in the two output waveguides.”

“The relation between the optical power and the phase signal is as follows, where ΔI is the power difference of the two outputs, and I_0 is the total power of the two outputs,

$$\phi \approx \sin(\phi) = \frac{\Delta I}{I_0} \equiv \Delta I_{MZI} .”$$

However, since we are comparing two testing results with the same total detected power, the data in Fig.3 and Fig.4 are not normalized. We added the sentence below on line 176 to clarify this.

“Since we are comparing the signals of our device and the regular MZI with equal detected optical power, the signals are not normalized.”

7. Line 143, the main body of the paper needs more detail about how the two devices, MZI and WVA are equivalent. This is in the SI but belongs here since the paper hinges on it.

We moved the content from supplementary information and added a paragraph on lines 158-177 in the testing section with description on how the two devices are equivalent.

“To confirm the enhancement of the signal, we also test an on-chip Mach-Zehnder interferometer with the same waveguide parameters and similar footprint with the same signal and testing setup. We also take the signal as the difference of the two waveguide outputs in the standard MZI. The relation between the optical power and the phase signal is as follows, where ΔI is the power difference of the two outputs, and I_0 is the total power of the two outputs,

$$\phi \approx \sin(\phi) = \frac{\Delta I}{I_0} \equiv \Delta I_{MZI} .$$

Comparing the power difference from the two devices with the same applied phase shift, we derive the amplification factor (details in supplementary information)

$$\frac{\Delta I_{WV}}{\Delta I_{MZI}} = \frac{0.45(1 - a)}{a} .$$

In our design, a is 0.125, which corresponds to a 9.97dB amplification of the signal. The standard MZI has the same arm length as the weak value device, except that one arm is shorter to put the MZI in quadrature. The waveguide and the heater in the MZI have the same dimensions as the single mode waveguide in the weak value device. Both heaters, including the contact probe and wires, have a resistance of $746 \pm 6 \Omega$. This guarantees that the same phase shift, within the error range, is applied to both devices when applying the same voltage to the heaters. We also test the standard MZI with the same setup as the weak value devices. We apply the same 10kHz sinusoid voltage wave to the heater and take the difference of the two outputs of the standard MZI with the balanced detector. Then we measure the signal with the spectrum analyzer using the same settings (100Hz span, 0.1Hz bandwidth). Since we are comparing the signals of our device and the regular MZI with equal detected optical power, the signals are not normalized.”

8. Fig. 3 refers to signals for the MZI and WVA but up to this point these signals have not been explicitly defined.

We added the sentences below on lines 152-157 to clarify how the signals are defined and measured.

“The optical outputs of the two waveguides are imaged on to a balanced detector with a 40x objective. The difference of the optical power on the two sides of the detector is converted to a voltage signal by the detector, which is then converted to electrical power on the RF spectrum analyzer. We take the electrical power signal as our signal and noise. The signal is measured with a resolution bandwidth of 0.1Hz.”

9. The authors should comment on the source of the noise in the system and its characteristics.

We added the sentences below on lines 196-201 to comment on the measurement and the source of the noise.

“The noise floors are measured with the RF spectrum analyzer at 20kHz over a span of 0.1kHz. They are the same for both devices because the same amount of optical power is received by the detector. At lower optical power levels ($<20\mu\text{W}$), the noise floor is limited by the built-in electronic amplifier noise and other electronic noise of the balanced detector. For detected power above $20\mu\text{W}$, the noise floor starts to grow with optical power, which indicates that the system starts to be shot noise limited.”

10. Fig 3 c. A horizontal axis for phase would be clearer than voltage.

We added a top axis to Fig.3c showing the corresponding phase.

c The signal of the weak value device and the regular MZI vs. heater voltage and phase change on a semi log plot. Signal and noise floor on weak value device (red circles) and standard MZI (blue squares) with equal detected optical power within 5% variation and changing applied heater signal (phase signal). The signal data are fitted to the lines. The top axis shows the corresponding phase shift introduced by the micro heater (details in supplementary information).

We also added the following section on how the phase is calculated with heat transfer simulations in the supplementary information on line 118.

“Micro heater and phase shift

Fig 6. **a** Temperature change vs. square of applied voltage on heater. **b** Effective index change vs. temperature change. **c** Phase change vs. applied voltage on heater.

We simulate and calculate the phase change with respect to the applied voltage on the micro heater. We first simulate the heat transfer of the structure with COMSOL and get the temperature at the waveguide. Then we use the thermo-optic coefficient of silicon nitride and silicon dioxide to simulate the effective index change of the waveguide mode. The refractive indices of silicon nitride and silicon dioxide increase linearly with temperature. The thermo-optic coefficients of the waveguide materials are $2.45 \times 10^{-5} \text{K}^{-1}$ for silicon nitride and $0.95 \times 10^{-5} \text{K}^{-1}$ for silicon dioxide¹. Finally, we calculate the propagation phase change with the following equation:

$$\Delta\phi = \Delta n_{eff} \frac{2\pi}{\lambda} * L_{heater} \quad (20)$$

where $\Delta\phi$ is the phase change, Δn_{eff} is the effective index change, λ is the wavelength, and L_{heater} is the length of the heater.

1. Arbabi, A. & Goddard, L. L. Measurements of the refractive indices and thermo-optic coefficients of Si_3N_4 and SiO_x using microring resonances. *Opt. Lett.*, OL **38**, 3878–3881 (2013).

11. Lines 159 to 165 start discussing results for “sensitivity” but it is not clear how they were found from the experiments nor defined theoretically.

In the paragraph starting from line 202 we changed “sensitivity” to “minimum resolvable signal”, which better describes our data. The minimum resolvable signal is obtained by extrapolating the fitted curve of signal till it crosses the noise floor. We added detected optical power and resolution bandwidth of the spectrum analyzer to clarify the conditions for the minimum resolvable signal.

We put in the following sentences lines 152-157, lines 196-201, and lines 204-205 to clarify our definition of the signal, the noise floor, and the minimum resolvable signal.

“The optical outputs of the two waveguides are imaged on to a balanced detector with a 40x objective. The difference of the optical power on the two sides of the detector is converted to a voltage signal by the detector, which is then converted to electrical power on the RF spectrum analyzer. We take the electrical power signal as our signal and noise. The signal is measured with a resolution bandwidth of 0.1Hz.”

“The noise floors are measured with the RF spectrum analyzer at 20kHz over a span of 0.1kHz. They are the same for both devices because the same amount of optical power is received by the detector. At lower optical power levels ($<20\mu\text{W}$), the noise floor is limited by the built-in electronic amplifier noise and other electronic noise of the balanced detector. For detected power above $20\mu\text{W}$, the noise floor starts to grow with optical power, which indicates that the system starts to be shot noise limited.”

“The minimum resolvable signal is obtained by extrapolating the fitted curve of the signal and the noise floor till they cross.”

12. Does the speed of modulation of the phase affect anything (e.g. the amount of noise)?

We added the sentence below on line 157 to comment on how the speed of modulation would affect the noise.

“At this modulation frequency, the contribution of the 1/f noise is reduced.”

13. The frequency modulation is confusing since there is now a modulation frequency and an optical frequency. E.g the sensitivity is stated as 2KHz, but of which frequency? The reader might find that confusing. Moreover, they are of the same size. Do the generated frequency sidebands from the sinusoidal modulation affect anything?

We added description and equation on lines 220-224 to clarify the different frequencies mentioned. 2kHz is of the optical frequency.

“We tune the laser wavelength on the slope of the resonance (Fig.4b, dashed line) and modulate the optical frequency with a sinusoid voltage signal of 20kHz (ω_m). The amplitude of the voltage signal determines modulation depth of the optical frequency (ω_d). The laser frequency (ω_{laser}) oscillates around the center frequency (ω_0) in the following manner:

$$\omega_{laser} = \omega_0 + \omega_d * \sin(\omega_m t).”$$

The side bands of the modulation have negligible effect on the devices. We added the following sentences on lines 225-228 to clarify this.

“The maximum modulation depth ω_d we apply on the optical frequency is 250kHz, which corresponds to 15fm of wavelength change. Such a laser wavelength change has negligible effect on the performance of the integrated photonic components, including the ring resonator (FWHM=2.5nm).”

14. Does the frequency shift consequentially alter the operation of the e.g. the MMI? If not, this should be stated with some brief justification.

No, the operation bandwidth of the components is much larger than the frequency shift. The maximum frequency shift we apply is 250kHz, which corresponds to 2fm of wavelength change. Such a wavelength change would result in an MMI power output difference change of 0.000006%. We added the following sentences on lines 225-228 to clarify this.

“The maximum modulation depth ω_d we apply on the optical frequency is 250kHz, which corresponds to 15fm of wavelength change. Such a laser wavelength change has negligible effect on the performance of the integrated photonic components, including the ring resonator (FWHM=2.5nm).”

15. In the SI, eq 12 is incorrect. I_0 should also be defined.

We corrected Eqn.12 and added definition of I_0 on lines 84-86 of supplementary information.

“For a standard MZI interferometer in quadrature, the phase shift is determined by the output power.

$$I = I_0 \left(\frac{1}{2} \pm \frac{1}{2} \sin(\phi) \right). \quad (12)$$

I_0 is the total input optical power and ϕ is the phase difference between the two paths.”

16. The SI is still not clear about the normalization in the WVA signal. Please define it with a formula.

We added the following section on lines 93-101 and lines 106-116 in supplementary information on how the weak value signal is normalized.

“To match the detected optical power to the MZI, the signal is normalized to the optical power in the dark port, which is also the input optical power into the MMI. We apply a linear fit of the simulation results to the following equation

$$p = c * (I_{MMI1} - I_{MMI2}) = c * \Delta I_{wv}$$

where p is the TE_0 field percentage in the input field, I_{MMI1} and I_{MMI2} are the power outputs of the two waveguides of the MMI. The difference between I_{MMI1} and I_{MMI2} is also the output of the entire weak value device, which we denote as ΔI_{wv} . The fitting parameter is $c = 1.11$.

In the simulation of the MMI outputs, we obtain its relation to the input TE_0 mode percentage ($TE_0:TE_1 = p : (1 - p)$).”

“We convert the MMI input mode parameter p (Eqn. 15) to phase shift and normalize the signal to total detected power. The optical power difference normalized to the total detected power of weak value device is

$$\Delta I_{wv} = \frac{0.9}{1 + \frac{2a}{\phi(1-a)}}.$$

For the same small phase shift ϕ , the ratio of weak value device output and regular MZI output is

$$\frac{\Delta I_{wv}}{\Delta I_{MZI}} = \frac{0.9}{\phi + \frac{2a}{(1-a)}}.$$

Since $\phi \ll a/(1-a)$,

$$\frac{\Delta I_{wv}}{\Delta I_{MZI}} = \frac{0.45(1-a)}{a}.$$

Comparing the optical power signal of the standard MZI and the weak value device, the calculated amplification is 3.15 in Fig.5, which corresponds to a 9.97dB increase of the spectrum analyzer signal.”

17. The SI should contain some theoretical justification for why the WVA will outperform the MZI. As a reminder, in the absence of technical noise, the two methods have almost equal sensitivity. If it is technical noise, the paper should explain the reason WVA outperforms the MZI for the type of technical noise observed for the chip should explained

We are not taking the input power as the limiting resource, but the detected power. The limitation of increasing signal-to-noise ratio is usually the power that can be detected before the detector reaches saturation. We added the sentence below on lines 46-47 with reference 22

“As detector saturation limits the maximum signal-to-noise ratio attainable, weak value amplification provides a further increase to the signal-to-noise ratio²².”

22. Starling, D. J., Dixon, P. B., Jordan, A. N. & Howell, J. C. Optimizing the signal-to-noise ratio of a beam-deflection measurement with interferometric weak values. Phys. Rev. A **80**, 041803 (2009).

and another sentence on line 192 to clarify this.

“We take the limiting resource as the maximum optical power that can be detected.”

We added a new preprint paper on arXiv as reference 23 on line 49 which contains the theory discussion of the comparison between weak value devices and MZI, as well as noise analysis.

“Weak value amplification techniques with optical interferometers have demonstrated improvement on sensitivity over traditional interferometers²³.”

23. Steinmetz, J., Lyons, K., Song, M., Cardenas, J. & Jordan, A. N. Enhanced on-chip frequency measurement using weak value amplification. arXiv:2103.15752 (2021).

Reviewer #2:

We thank the reviewer for a careful reading of the paper and giving insightful comments of our results. We now reply point by point:

“However, focusing on the translation of a well-known and well understood effect to a different platform, I am not sure whether the present manuscript offers quite enough novelty and innovation to influence thinking in the field.”

Bringing weak value technique onto an integrated photonic platform opens many technological applications in on-chip sensing, optical communications, and quantum optical technologies. Theoretically, our method of using mode expansion to interpret weak value amplification fields is also key to bridging the gap between free space and integrated photonics and brings new insight into the fundamental physics of this effect. Practically, integrated photonics provides the advantages of stabilization and miniaturization of the interferometry in weak value setups.

“Moreover, in my opinion the presentational standard of the current manuscript needs some work, for several reasons listed below and as follows: probably in an attempt to make it suitable for a broad readership, it provides a verbal and non-technical description which in my opinion lacks depth even for the non-specialist reader. In fact, the absence of equations and a more quantitative description makes it hard to follow in detail and appreciate the work fully for the expert reader as well.”

We added equations in the main text to make a more quantitative description.

From line 96, “The electric field at the dark port can be expressed by the following equation, where ϕ is the phase difference between the two paths and a is the amplitude of the TE_0 mode that couples to the TE_1 mode (details in supplementary information).

$$E_d \approx i \left[(1 - a) \frac{\phi}{2} TE_0 + a TE_1 \right] = ia \left[TE_1 + \frac{1-a}{a} \frac{\phi}{2} TE_0 \right].”$$

From line 137, “From the power difference, we can get the amplitude ratio $TE_0:TE_1 = p/(1 - p)$, where $p = 1.11\Delta I_{wv}$ for this design (details in supplementary information). ΔI_{wv} is the power difference normalized to the total power in the two output waveguides. Then the phase signal can be obtained by (details in supplementary information)

$$\phi = 2 \frac{p}{1-p} \frac{a}{1-a}.”$$

From line 160, “The relation between the optical power and the phase signal is as follows, where ΔI is the power difference of the two outputs, and I_0 is the total power of the two outputs.

$$\phi \approx \sin(\phi) = \frac{\Delta I}{I_0} \equiv \Delta I_{MZI} .$$

Comparing the power difference from the two devices with same applied phase shift, we derive the amplification factor (details in supplementary information)

$$\frac{\Delta I_{wv}}{\Delta I_{MZI}} = \frac{0.45(1-a)}{a} . , ,$$

- The authors write about it being desirable to operate at the shot noise limit as a major motivation for weak value amplification. While it is true this represents the ultimate limit of sensitivity on a per photon basis, it is not immediately obvious to me that working in this limit of low signal intensity is always desirable, and perhaps this should be discussed and explained in more detail.

We thank the referee for this comment. The critical point is what one takes to be the limiting resource in a metrology experiment. In our situation, it is the maximum power the detector can receive, rather than the power of the laser.

Our goal is not to work with low signal intensity but to further bring up the signal by increasing the laser power, so that the detector is saturated, and the signal cannot be increased by increasing the optical power anymore. The comparison to the standard MZI is then made with equal amount of detected power. We added the sentence below on lines 46-47 with reference 22

“As detector saturation limits the maximum signal-to-noise ratio attainable, weak value amplification provides a further increase to the signal-to-noise ratio²².”

22. Starling, D. J., Dixon, P. B., Jordan, A. N. & Howell, J. C. Optimizing the signal-to-noise ratio of a beam-deflection measurement with interferometric weak values. Phys. Rev. A **80**, 041803 (2009).

and this sentence on line 192 to clarify this.

“We take the limiting resource as the maximum optical power that can be detected.”

- Weak value amplification can certainly be valuable and useful, as demonstrated by many impressive experimental results over the years. However, it comes at a cost of low post-selection probability, and there are basic arguments that post-selection should never be useful at a fundamental level: why not simply keep and process all the data? I recall an interesting debate from a while ago about whether and under what conditions weak value amplification helps. I appreciate the authors may consider this a bit peripheral and may not wish to discuss this in too much detail (and I agree the specific advantages they describe all seem valid), however, for a more balanced introduction, I would nonetheless have expected some degree of coverage of these issues.

We thank the referee for this comment. Indeed, over the years, there has been extensive debate over this topic. However, we view that this debate has been conclusively resolved in both theoretical and experimental findings. The point is that while it is true that some information about the desired parameter still resides in the rejected events, a properly designed weak value experiment can make this information a negligible fraction of the total, effectively concentrating all the available information into a tiny fraction of events.

We added the sentences below on lines 44-47 with reference 10, 20, 21, and 22 to comment on this.

“While there is some information in the rejected port (the bright port), it is a negligible fraction of the total information^{10,20,21}. As detector saturation limits the maximum signal-to-noise ratio attainable, weak value amplification provides a further increase to the signal-to-noise ratio²².”

10. Jordan, A. N., Martínez-Rincón, J. & Howell, J. C. Technical Advantages for Weak-Value Amplification: When Less Is More. *Phys. Rev. X* **4**, 011031 (2014).

20. Viza, G. I., Martínez-Rincón, J., Alves, G. B., Jordan, A. N. & Howell, J. C. Experimentally quantifying the advantages of weak-value-based metrology. *Phys. Rev. A* **92**, 032127 (2015).

21. Sinclair, J., Hallaji, M., Steinberg, A. M., Tollaksen, J. & Jordan, A. N. Weak-value amplification and optimal parameter estimation in the presence of correlated noise. *Phys. Rev. A* **96**, 052128 (2017).

22. Starling, D. J., Dixon, P. B., Jordan, A. N. & Howell, J. C. Optimizing the signal-to-noise ratio of a beam-deflection measurement with interferometric weak values. *Phys. Rev. A* **80**, 041803 (2009).

We added a new preprint paper on arXiv as reference 23 on line 49 which contains theory discussion of the comparison between weak value devices and MZI, as well as noise analysis.

“Weak value amplification techniques with optical interferometers have demonstrated improvement on sensitivity over traditional interferometers²³.”

23. Steinmetz, J., Lyons, K., Song, M., Cardenas, J. & Jordan, A. N. Enhanced on-chip frequency measurement using weak value amplification. arXiv:2103.15752 (2021).

- I found the sentence spanning lines 57-59 hard to read, this could be rephrased and possibly expanded upon.

We modified the sentences on lines 54-60 to make them clearer.

“Weak value amplification (WVA) consists in measuring the spatial phase front tilt, using the known phase shift to amplify the signal⁸. On the other hand, inverse weak value amplification (IWVA) consists in measuring the phase shift with the signal amplified by the known spatial phase front tilt²⁴. In the WVA regime, the measured parameter, phase front tilt, is smaller than the propagation phase shift. Meanwhile, in the IWVA regime, the propagation phase shift is smaller than the phase front tilt, which is opposite from WVA. The two operating regimes allow different applications of weak value techniques.”

- There seems to be a grammar issue in the sentence spanning lines 73-74, possibly just a 'to' missing.

We corrected the sentence on lines 76-77.

“Then light in each arm goes through a spatial phase front tilter and undergoes an opposite spatial phase front tilt. They interfere at another multi-mode 50/50 directional coupler.”

- In Fig.1, I'm not sure how helpful it is to invoke the Sagnac interferometer in (b), seeing the implemented device is a MZ, as shown in (a). I appreciate the Sagnac example forms the basis of discussion in the introduction, but should this not ideally be done with reference to an the MZ setup to help the reader understand the experimental device?

We use the Sagnac interferometer as a base line to explain our mode expansion theory because of its wide adoption in previous works, which would be more accessible to the readers. Also, the Sagnac interferometer is not essentially different from the MZI in theory. For the integrated device, we put the full derivation in the supplementary information to help the readers to understand the theory.

- The quality and polish of the figures could be improved in terms of appearance and resolution.

We increased the resolution of the figures. We replaced figures with vector images as much as possible and included them in the submission.

- The Supplementary Materials document contains a number of typos and would benefit from thorough proofreading. In my view, it is not possible to properly understand the phase tilt implementation from the main text without reading the helpful and fairly easy to follow derivation in the SM. I am not sure if the split between those documents is quite ideal.

We have moved some equations from the supplementary information to the main text as well as fixed typos and carefully proofread the supplementary information.

- The "standard" MZ benchmark seems fair and carefully set up. To assess the potential and usefulness of this approach for practical applications, some level of discussion of how this integrated WVA compares to the performance of possible competing approaches more widely than the implemented MZI would be valuable. While admittedly an apples to oranges comparison, I would also find it interesting how it compares to best-of-class / typical table top interferometers, should the authors have relevant data available.

To try to make an apples-to-apples comparison, we compared our device to the free space frequency measurement with inverse weak value amplification (reference 26) on lines 236-240.

“Previous free space frequency measurements with inverse weak value amplification show a $129\text{kHz}/\sqrt{\text{Hz}}$ sensitivity (i.e. an optical frequency shift of 129kHz could be measured with an integration time of 1sec and an SNR of 1)²⁶. Taking into consideration the resolution bandwidth of 0.1Hz , our 2kHz detection limit corresponds to a sensitivity of $6.3\text{kHz}/\sqrt{\text{Hz}}$.”

26. Starling, D. J., Dixon, P. B., Jordan, A. N. & Howell, J. C. Precision frequency measurements with interferometric weak values. *Phys. Rev. A* **82**, 063822 (2010).

Sincerely,

The Authors.

REVIEWERS' COMMENTS

Reviewer #1 (Remarks to the Author):

The authors have satisfactorily responded to my concerns about the manuscript. In particular, they have clarified what they mean by signal, noise, and sensitivity. Moreover, they have clearly defined the sense in which their inverse weak value amplification device outperforms the standard device.

As far as I know, all weak value amplification experiments that have shown some sort of advantage have been proof-of-principle demonstrations. Past demonstrations have been somewhat contrived and were narrowly focused on demonstrating the amplification effect in some regime, regardless of whether it was a practical operating regime or not. The value of this work is that it shows that the concept works in practice (or, at least, closer to practice).

I agree with the other reviewer that the paper does not lead to deep insights or understandings. Nonetheless, quantum-enhanced or motivated metrology has had few practical demonstrations (gravitational wave detection and cold-atom atomic clocks being exceptions). Almost every paper, including dozens in Nature journals, have resorted to proof-of-principle demonstrations. Most don't even show a true enhancement in sensitivity. So, in this sense, this is a landmark paper that should be published in Nature Communications.

Reviewer #2 (Remarks to the Author):

I appreciate the authors' detailed response as well as their efforts to revise their manuscript. I am happy with their replies (both to myself as well as to the many insightful and valid questions raised by the other reviewer). In my view, the revised version is much improved, and in particular more specific about what exactly has been done and how it has been accomplished. I am still not sure how accessible it is to a broad readership, and whether the balance that has been struck between (quantitative, i.e. equation based) background on WVA and IWVA and the device side of things is quite optimal, or whether invoking both Sagnac and Mach-Zehnder interferometers in the main text is the best way to present the story. In any case, this revision is a big step in the right direction but, in my opinion, the authors could have gone further and restructured the manuscript more radically. However, this may be down to personal preferences, and I suppose the current focus is in keeping with highlighting the key advance of the paper.

Regarding my previous concern about this work constituting more of a technical / practical advance (as opposed to conceptual / scientific), I find the authors' reply helpful and convincing. In combination with Editorial advice (in the reviewer request communication to me), I am satisfied that this manuscript reports important results that represent a significant step towards more practical devices for quantum optical metrology, and that this work is thus congruent with publication in Nature Communications.

A couple of minor issues remain, it would be good if these could be addressed or clarified prior to publication:

- whilst the authors are now sufficiently explicit about the detected power being the limited resource (and my reading is the total dark port power in the IWVA approach matches the combined output power of both ports for the MZI - is this correct?), I would find it valuable to also have the respective input power stated and compared for both cases. This would provide additional context and a different perspective on the involved resource.
- I'm afraid I do not quite follow the sentence on lines 200 / 201 and how this can be reconciled with Fig 3a. Perhaps the authors could elaborate?
- Regarding the text in line 207 that the MZI can resolve a voltage of $1.2 \mu\text{rad}$, how does this relate to Fig 3b? It seems the resolvable signal is smaller than that (but the also stated 0.02 V looks right). Am I misinterpreting this, or is it a conversion error?

Reviewer 2:

“A couple of minor issues remain, it would be good if these could be addressed or clarified prior to publication:

- whilst the authors are now sufficiently explicit about the detected power being the limited resource (and my reading is the total dark port power in the IWVA approach matches the combined output power of both ports for the MZI - is this correct?), I would find it valuable to also have the respective input power stated and compared for both cases. This would provide additional context and a different perspective on the involved resource.”

Yes, the total dark port power in the IWVA device matches the combined output power of both ports of the standard MZI. We put the sentence below on lines 201-203 to clarify how much laser power is used in both devices:

“For a detected power of $130\mu\text{W}$, the weak value device has an input laser power of 16mW and the standard MZI has an input laser power of 0.5mW .”

“- I'm afraid I do not quite follow the sentence on lines 200 / 201 and how this can be reconciled with Fig 3a. Perhaps the authors could elaborate?”

We modified the sentences on lines 206-208 to clarify this.

“For detected power above $20\mu\text{W}$, the noise floors of both devices start to grow linearly with optical power in the log scale plot, which indicates that the systems start to be shot noise limited.”

“- Regarding the text in line 207 that the MZI can resolve a voltage of $1.2\ \mu\text{rad}$, how does this relate to Fig 3b? It seems the resolvable signal is small than that (but the also stated $0.02\ \text{V}$ looks right). Am I misinterpreting this, or is it a conversion error?”

We thank the reviewer for noticing the error. The voltage should be 0.012V instead of 0.02V . We also changed the phase shift correspondingly. On lines 213-214, we corrected the sentence:

“The standard MZI shows a minimum resolvable signal of 0.012V , which corresponds to a phase signal of $0.44\mu\text{rad}$ (details in Supplementary Note 5).”

Sincerely,

The Authors